# The Methylation Patterns and Transcriptional Responses to Chilling Stress at the Seedling Stage in Rice

**DOI:** 10.3390/ijms20205089

**Published:** 2019-10-14

**Authors:** Hui Guo, Tingkai Wu, Shuxing Li, Qiang He, Zhanlie Yang, Wuhan Zhang, Yu Gan, Pingyong Sun, Guanlun Xiang, Hongyu Zhang, Huafeng Deng

**Affiliations:** 1State Key Laboratory of Hybrid Rice, Longping Branch of Graduate School, Central South University, Changsha 410013, China; nksgh2008@163.com; 2Rice Research Institute, Guizhou Academy of Agriculture Sciences, Guiyang 550006, Chinagzrice@126.com (G.X.); 3Rice Research Institute, Sichuan Agricultural University, Chengdu 611130, China; 4Hunan Hybrid Rice Research Center, Hunan Academy of Agricultural Sciences, Changsha 410125, China

**Keywords:** cold stress, transcriptome, RNA-Seq, methylation, rice (*Oryza sativa* L.)

## Abstract

Chilling stress is considered the major abiotic stress affecting the growth, development, and yield of rice. To understand the transcriptomic responses and methylation regulation of rice in response to chilling stress, we analyzed a cold-tolerant variety of rice (*Oryza sativa* L. cv. P427). The physiological properties, transcriptome, and methylation of cold-tolerant P427 seedlings under low-temperature stress (2–3 °C) were investigated. We found that P427 exhibited enhanced tolerance to low temperature, likely via increasing antioxidant enzyme activity and promoting the accumulation of abscisic acid (ABA). The Methylated DNA Immunoprecipitation Sequencing (MeDIP-seq) data showed that the number of methylation-altered genes was highest in P427 (5496) and slightly lower in Nipponbare (Nip) and 9311 (4528 and 3341, respectively), and only 2.7% (292) of methylation genes were detected as common differentially methylated genes (DMGs) related to cold tolerance in the three varieties. Transcriptome analyses revealed that 1654 genes had specifically altered expression in P427 under cold stress. These genes mainly belonged to transcription factor families, such as Myeloblastosis (MYB), APETALA2/ethylene-responsive element binding proteins (AP2-EREBP), NAM-ATAF-CUC (NAC) and WRKY. Fifty-one genes showed simultaneous methylation and expression level changes. Quantitative RT-PCR (qRT-PCR) results showed that genes involved in the ICE (inducer of CBF expression)-CBF (C-repeat binding factor)—COR (cold-regulated) pathway were highly expressed under cold stress, including the WRKY genes. The homologous gene *Os03g0610900* of the open stomatal 1 (OST1) in rice was obtained by evolutionary tree analysis. Methylation in *Os03g0610900* gene promoter region decreased, and the expression level of *Os03g0610900* increased, suggesting that cold stress may lead to demethylation and increased gene expression of *Os03g0610900*. The ICE-CBF-COR pathway plays a vital role in the cold tolerance of the rice cultivar P427. Overall, this study demonstrates the differences in methylation and gene expression levels of P427 in response to low-temperature stress, providing a foundation for further investigations of the relationship between environmental stress, DNA methylation, and gene expression in rice.

## 1. Introduction

Low temperature is one of the main environmental factors that limits plant growth, development, crop yield, and geographical distribution, and is also a natural disaster encountered in crop production. To adapt to low-temperature stress, plants have developed complex and efficient response mechanisms at the molecular, cellular, physiological, and biochemical levels through long-term evolutionary processes. Low temperature disrupts plant metabolism by inhibiting metabolic enzymes or directly altering gene expression [1]. 

Most temperate plants acquire freezing tolerance by a process called cold acclimation by exposure to nonfreezing temperatures for a period of time [2]. Upon exposure to low temperatures, the expression of Cold-Regulated (COR) genes generates protective proteins, including detoxifying enzymes, key proteins in osmotic biosynthesis, fatty acids, and antifreeze proteins, which protect cells from frostbite [2]. C-repeat binding factors (CBFs) and dehydration response element binding protein (DREB1S) are the major transcription factors involved in cold acclimation. The corresponding transcripts are rapidly induced after cold exposure. CBFs and DREB1S directly regulate the expression of the downstream COR genes (also known as the CBF regulator) and modulate cold stress [2,3].

In plants, a well-known transcriptional regulatory pathway involved in plant cold adaptation is the CBF/DREB1 cold signaling pathway mediated by CBF transcription factors. Signal transduction occurs via the ICE-CBF-COR cascade [1]. In this pathway, CBF1 and CBF3 are two key CBF transcription factors involved in the plant low-temperature stress response. Previous studies have shown that overexpression of *CFL1* and *CBF3* significantly improves the cold tolerance of plants, while their mutants have significantly reduced cold tolerance [4,5]. ICE1 (inducer of CBF expression 1) is an important upstream transcription factor. Under cold stress, ICE1, encoding a MYC-type basic helix-loop-helix (*bHLH*) transcription factor, directly binds to the promoters of CBF genes through the *MYC* recognition site and then activates the transcription of CBF3/DREB1 genes. The transcripts bind to the CRT/DRE core elements of the cold-responsive gene promoter region and activate transcription of the COR gene, subsequently triggering a series of anti-cold physiological and biochemical reactions [6]. Studies have shown that overexpression of the *Arabidopsis ICE1* gene in rice increases the cold tolerance of rice [7]. *ICE1* is a constitutively expressed gene with different gene expression levels in different cold tolerance varieties; however, its differential expression regulatory mechanism is still unclear [8].

A previous publication has reported that open stomatal 1 (OST1) plays a key role in plant freeze resistance [9]. Cold-activated OST1 interacts with and phosphorylates ICE1 to avoid ubiquitination degradation of the ICE1 protein, thereby enhancing the plant’s ability to tolerate freezing [9]. Yang et al. (2018) found that BTF3 and BTF3L are OST1 interaction proteins [10]. Under low-temperature conditions, BTF3L is phosphorylated by the activated OST1 protein, thereby enhancing its interaction with CBFs. As a result, the stability of CBF protein is enhanced, the downstream low-temperature response gene COR is highly expressed, and the plant’s antifreeze ability is enhanced. Furthermore, MPK3/6 negatively regulates the stability of ICE1 protein, resulting in reduced plant freezing tolerance in *Arabidopsis* [11,12]. However, *OsMPK3* positively regulates rice cold tolerance by inhibiting *OsICE1* degradation under cold stress [13]. Recently, plasma membrane-localized cold-response protein kinase 1 (CRPK1) was found to negatively regulate plant freezing tolerance. The phosphorylated 14-3-3 protein mediated by CRPK1 shuttles from the cytoplasm to the nucleus and interacts with the CBF protein, thereby destabilizing the CBF protein under cold stress [14].

Zhang [15] et al. performed a comparative microarray analysis of rice varieties that are cold-tolerant (LTH, Japonica) and cold-sensitive (IR29, Indica). The results showed similar early responses of the two varieties to low temperature, but the expression of genes with different functional categories later point. 

DNA methylation, one of the most important epigenetic marks in plants, plays an important role in the regulation of gene expression and cellular differentiation [16,17,18]. DNA methylation refers to the addition of a methyl group to the 5’ carbon atom of the cytosine molecule of DNA by thiomethionine as a methyl donor catalyzed by DNA methyltransferase [19]. Under stress conditions, DNA methylation can regulate the expression of stress response genes and plant growth [1]. DNA methylation levels are affected by abiotic stresses, such as low temperature, salt and drought [20,21,22,23,24,25]. The change in DNA methylation level is characterized by stable inheritance [26,27]. The cold tolerance of *Ageratina adenophora* is the main mechanism responsible for its invasion and spreading to the north of China. The increase in cold tolerance of the population is caused by the up-regulated expression of the ICE-CBF transcription pathway. Among the CBF/DREB1-dependent cold signaling pathways, only the *ICE1* gene showed methylation changes among the *Ageratina adenophora* populations, and methylation was significantly correlated with gene expression and cold tolerance [8].

To expand our understanding of the complex mechanisms involved in cold tolerance in rice, we screened many breeding materials for low-temperature tolerance. A rice cold-tolerant variety, P427, was identified via low-temperature treatment (2–3 °C). The DNA methylation mapping and expression profiling were used to reveal the pathway involved in cold signaling transduction of P427 under cold stress. The homologous gene *Os03g0610900* in rice was identified by aligning with the OST1 gene of *Arabidopsis thaliana*. The cold-tolerant biological function of this gene has not yet been reported. This study found that these homologous genes are involved in the signaling pathway of ABA in cold stress. We identified 3 other genes related to cold tolerance. Our study showed that the cold-tolerant variety P427 improves plant cold tolerance by positively regulating the signaling pathway of OST1-ICE1 during cold stress. These results reveal the function of *Os03g0610900* in cold adaptation and the potential of the P427 allele in molecular breeding in rice.

## 2. Results

### 2.1. The Effects of Low-Temperature Stress on Seed Germination, Growth, and Survival of Three Rice Cultivars Seedlings

Through the screening of local resource varieties, we identified a low-temperature-resistant variety, P427. To analyze the molecular mechanism of chilling stress tolerance, the cold-tolerant variety P427, a Japonica rice variety, Nipponbare (Nip), and an Indica rice variety, 9311 were used to compare seedling stage of rice. We first investigated the seed germination rate and seedling growth under cold conditions. When the seeds germinated to approximately 5 mm long under normal conditions, low-temperature (2–3 °C) treatment was administered for 3 days, and then the seedlings were transferred to normal conditions for 2 weeks. The seedling length of rice was not different before and after treatment among the three materials (Figure 1A); however, the survival rate of P427 was 88%, which was much higher than the other two varieties (Appendix A), indicating a significant effect on late growth and development. These results also showed strong cold resistance of P427 at the germination stage.

To confirm that P427 had a higher degree of cold tolerance than Nip and 9311 at the seedling stage, 3-week-old seedlings were subjected to low-temperature treatment at 2–3 °C for 3 days and then transferred to normal growth conditions for 7 days for recovery. The leaves of Nip and 9311 displayed curling and wilting after cold treatment, which are signs of early programmed cell death (PCD) in the leaf, causing etiolated and withered leaf formation (Figure 1B). However, these was no significant difference in leaf color of P427 before and after treatment, and the leaves only partially underwent PCD during recovery. These results confirmed that P427 was more cold-tolerant than Nip and 9311.

To further characterize the cold tolerance of these three cultivars, we measured the chlorophyll content in the leaves. The seedlings were exposed to the dark conditions for 1 h, followed by immediate measurement of the chlorophyll content. Then, the seedling was transferred to 2–3 °C for 3 days and subsequent recovery for 3 days. The chlorophyll content in the leaves was measured at each time point. We observed that the physiological condition of all plants was healthy before cold treatment. After 3 days of cold treatment, the chlorophyll content of Nip decreased by 10.88%, and the chlorophyll content of 9311 decreased by 14.77%, which is consistent with the previous finding that Indica rice is more sensitive to temperature than Japonica rice. The chlorophyll content of P427 increased by 0.2% (Figure 1C). After 3 days of recovery growth, the chlorophyll content of the three materials decreased (9311: 21.24; Nip: 23.1; P427: 27.96) (Figure 1C). These results indicate that P427 shows low sensitivity to cold stress and significantly higher cold tolerance than Nip and 9311.

### 2.2. Physiological Response of Seedlings to Low Temperature

Low temperature or other stresses increase the levels of intracellular reactive oxygen species (ROS) and trigger lipid peroxidation in the cell membrane. To adapt to these stresses, plants have evolved self-protection mechanisms, i.e., enzymatic and nonenzymatic systems. We analyzed the relationships of hydrogen peroxide (H_2_O_2_), abscisic acid (ABA) content, superoxide anion radical (O_2_•^−^), catalase (CAT), peroxidase (POD) and superoxide dismutase (SOD) activity after cold stress. The results showed that these six indicators were up-regulated during the low-temperature treatment, peaking at 72 h after the end of the low-temperature treatment. During the recovery period, these indicators began to decline. The experimental data showed stronger enzymatic activity of P427 than Nip and 9311. While the ABA content in all varieties increased during low-temperature treatment and decreased during recovery growth, P427 showed a more rapid rate of increase. The changes in the patterns of H_2_O_2_ were different. During the whole low-temperature treatment, 9311 showed a faster rate of increase. The H_2_O_2_ content in P427 and Nip also increased but relatively slowly, showing a similar pattern of change. After 24 h of recovery under normal growth conditions, the final content or activity of the six physiological indicators tended to be consistent (Figure 2). Free radicals will be transformed to H_2_O_2_ by SOD in rice under stress conditions. Then, CAT will decompose H_2_O_2_ into oxygen and water to protect the rice from the toxicity of H_2_O_2_. In our study, we found that the contents of SOD, CAT, and POD increased and were higher in P427 than in other two varieties. Additionally, the H_2_O_2_ content of P427 was the lowest among the three varieties. Thus, the cells of P427 possess better redox balance abilities with advantages in plant defense, stress responses, and delayed senescence under abiotic stress.

### 2.3. Comparison of DNA Methylation Levels and Loci 

To test whether low-temperature stress affects the change in genome structure and to identify possible mechanisms involved in chilling stress, Methylated DNA Immunoprecipitation Sequencing (MeDIP-seq) was used to analyze the DNA methylation distribution in the genomes of Nip, 9311, and P427 under cold stress. We generated a total of 5.2–6.8 million sequence reads. Of the total clean reads from the three samples, more than 51.12% were uniquely aligned to the rice genome in a single sample (Appendix A). We mainly calculated the proportion of DNA methylation at the cytosine site, and we found a lower cytosine DNA methylation rate of P427 and 9311 under cold stress compared with the rate under normal temperature, while the Nip cytosine DNA methylation rate increased slightly under cold stress. The three methylation in different contexts, CG, CHG, and CHH, were similar in proportion (CG > CHH > CHG) among the three materials of P427, Nip, and 9311. Under cold stress, the proportion of CG and CHG increased, while CHH decreased. Among the three materials, the most significant changes were observed for CG (+1.33%) and CHH (−2.09%) of P427. Together, these data showed that the decrease in CHH in rice after cold stress might be helpful in improving their cold resistance. 

To identify genes with significant differences in DNA methylation levels before and after cold stress, the genes with a read abundance ≥ 2-fold and with a *p*-value < 0.05 were defined as differentially methylated genes (DMGs). A Venn diagram shows the overlap of these DMGs among the three samples (Figure 3A). The results showed that the number of methylation-altered genes was highest in P427 (5496) and slightly less in Nip and 9311 (4528 and 3341, respectively), and only 2.7% (292) genes were detected as common DMGs related to cold tolerance in the three materials. Among these DMGs, 33.2% (3541) were unique to P427, indicating that the altered methylation levels of these genes likely contributed to the resistance of P427 to cold stress. 

We further examined the proportion of methylated CG, CHG, and CHH contexts in the promoter and gene body regions of the three materials under cold stress (Figure 3B). The results showed that the most detected sites were present in P427 (6230), while 9311 had the fewest detected sites (3612). However, a higher proportion of CHH sites was found in 9311 (48%), suggesting that sub-specification may have contributed to the methylation difference. The main explanation for this possibility is that 9311 is an Indica rice, and P427 and Nip are Japonica rices. It is speculated that the high proportion of CHH loci in 9311 was due to the difference between Indica rice and Japonica rice subspecies. 

We further analyzed the changes in the methylation levels of 292 shared cold-sensitive genes in these three materials (Appendix A). The results showed that the DNA methylation levels of P427 and 9311 were consistent, i.e., the number of genes with decreased DNA methylation was higher than the increased number. The trend in the number of DNA methylation distributions in different regions was intron > promoter > exon > 3′UTR > 5′UTR > EST. Additionally, an analysis of the proportion of the three methylation patterns (CG, CHG, CHH) in the promoter region showed that the proportion of CHG in the three materials was quite different, i.e., the proportion in Nip was as high as 51.4%, and the lowest proportion in the 9311 was 31.61%. Furthermore, we found that CG had the lowest proportion in Nip (40, 22.35%), with no significant difference (Appendix A). These results indicated that the 292 cold-sensitive genes did not play a major role in resistance to cold stress. 

### 2.4. Functional Enrichment Analysis of the Methylation Differential Gene of P427

We focused on the cold tolerance of P427. For the 5496 hypermethylated genes in P427 shown in Figure 3A, we analyzed the ratio of the three methylation types (CG/CHG/CHH) in the promoters and gene body regions (Appendix A). We found that the proportion of CG types was highest (54%, 44%), and the proportions of CHG and CHH were not very different. Therefore, we concluded that the hypermethylation of the CG type in promoter and gene body regions after cold stress treatment could inhibit the expression of certain genes and improve the cold tolerance of plants.

To obtain more functional information, we performed a high methylation screen of 3541 genes in P427 (|log2FC| ≥ 2 and *p*-value ≤ 0.01) using the gene ontology (GO) database (http://bioinfo.cau.edu.cn/agriGO/). We found that GO terms were associated with 1414 genes and were classified into 30 functional subcategory annotations (Figure 4). Molecular function made up most of the GO annotations, followed by biological processes and cellular component. Specifically, the significantly enriched GO terms in the molecular function categories were associated with monooxygenase activity, oxidoreductase activity, and iron ion binding. Taken together, these data indicated that cold stress activated plant redox activities, which protected the plant by reducing the content of H_2_O_2_. 

### 2.5. Comparison of mRNA Expression Differences Under Low-Temperature Stress

To further analyze and explore the effects of cold stress on the expression and regulation of genes in different materials, RNA sequencing (RNA-Seq) was performed for genome-wide gene expression profiling to compare the three varieties under cold stress. Total RNAs were extracted from leaf samples of P427, Nip, and 9311, which were treated at 3 °C for 3 days. Based on the RNA-Seq data, we used log_2_FC ≥ 1 or log_2_FC ≤ 0.25, with a *p*-value ≤ 0.01, as the screening cut-off, and the results revealed 6864 shared differentially expressed genes (DEGs) and 1654 P427-specific DEGs (Figure 5).

Transcription factors play an important role in the cold tolerance signaling pathway. To further identify transcription factors involved in chilling tolerance responses in rice, we examined all the transcription factors predicted from the database among the unique 1654 DEGs in P427 (Appendix A). A total of 67 family subcategories were identified, of which four family subcategories, the MYB family (68 genes), the AP2-EREBP family (63 genes), the NAC family (48 genes), and the WRKY family (46 genes), were included in the top four most abundant subgroups. The low-temperature response of plants is a complex process that involves multiple genes and multiple signaling pathways. Figure 5 depicts transcriptional regulators of the low-temperature response of P427 during noncold domestication. These results expand our understanding of the complex transcriptional regulation involved in chilling tolerance in rice.

### 2.6. Combined Analysis of The Methylation and Expression Profiles of Unique Genes in P427

To explore whether the changes in DNA methylation were related to the gene expression in P427, we compared the unique genes (1244) with significant changes in methylation levels in the promoter region before and after cold treatment with the 1654 differentially expressed genes from the expression profile. The Venn diagram revealed only 51 genes with changes in methylation and expression levels (Figure 6), 22 genes with a positive correlation between methylation and expression, and 29 genes with a negative correlation (Appendix A). These 51 genes mainly undergo methylation in the CG context, mostly occurring in the promoter region of the gene (Appendix A).

Most of these 51 genes were methylated in CG mode, and most of these methylations occurred in promoter regions. The log2FoldChange expression level was greater than +5, the corresponding methylation level increased, and the change pattern was mainly CG. Analysis of down methylated genes showed that CHH was the most common methylation mode, followed by CG. These results further confirmed that the increase in CG methylation and decrease in CHH methylation were more conducive to improving the cold resistance of plants. Of the 51 genes, 44 (86.27%) genes showed increased methylation levels, while 30 (58.82%) genes showed decreased expression (Appendix A). LI [28] and Lister R [29] found that methylation in the promoter could inhibit gene transcription. The results of this study are consistent with previous observations, indicating that in response to cold stress, the methylation levels generally increased in promoter regions of hypermethylated genes in P427.

Due to the large changes in their expression levels, GO enrichment analysis was performed to clarify the biological function of the 1603 expressed genes in P427 (Figure 7). The results showed that the functions of these genes were enriched in three main categories: cellular component, biological process, and molecular function. The significantly enriched GO terms of cellular component were associated with intracellular organelles (412, 25.7%) and intracellular membrane-bounded organelles (382, 23.83%) (Figure 7A). We also performed an enrichment analysis of the Kyoto Encyclopedia of Genes and Genomes(KEGG) pathway to examine the metabolic regulation of these 1603 genes (Figure 7B). The results showed that the genes were mainly enriched in metabolic pathways (94 genes), most of which were distributed among biosynthesis of secondary metabolites (60, 63.83%), with a small number distributed in plant hormone signal transduction, Phenylpropanoid, Biosynthesis, and Purine metabolism, among others.

### 2.7. Validation of Gene Expression Profiles by qRT-PCR

To verify the DEG analysis using transcriptome data, we validated the expression patterns of 10 randomly selected genes (5 genes in the ICE-CBF-COR pathway, 2 other cold-tolerance-related genes, and 3 unknown new genes) using qRT-PCR. Total RNA was extracted from the leaves obtained from the three materials after stress treatment (Figure 8). The results demonstrated that the quantitative expression patterns of these 10 genes were consistent with the results of the RNA-Seq analysis. We found that the expression levels of the five genes in the ICE-CBF-COR pathway were significantly increased after cold stress treatment, with the highest levels detected in P427, indicating that the ICE-CBF-COR pathway might play an important role in improving plant cold tolerance. WRKY7 and WRKY70 also showed increased expression under cold stress, indicating that cold tolerance caused widespread expression changes in related anti-cold genes. We also identified several new genes of unknown function that were expressed at high levels, three of which are listed in this article. Among these genes, *Os06g0638900* and *Os04g0497000* showed increases in expression levels after cold stress, while the quantitative RT results for *Os04g0463200* yielded results that were opposite to those of the DEG analysis. Aakash Chawade [30] also found that *Os04g0497000* was expressed at high levels after cold stress in Nepalese plateau Japonica rice. However, whether the cold tolerance of P427 requires the contribution of these three new genes requires further experiments.

*OST1* is an important gene in the ICE-CBF-COR pathway, which can enhance the frost resistance of plants by phosphorylating ICE1 [9]. OST1 can also improve the low-temperature tolerance of plants by increasing COR expression by phosphorylation of BTF3. Previous studies on the dicotyledonous model plant *Arabidopsis thaliana* have described the ICE-CBF-COR cold stress regulation signaling pathway. However, this signaling pathway has not been clearly proposed and revealed in monocotyledonous rice. In this study, transcriptome sequencing analysis revealed that rice genes homologous to the ICE-CBF-COR signaling pathway in *Arabidopsis thaliana* were enriched and differentially expressed (Figure 9). In rice, most of the direct homologous genes in this pathway have been identified to have physiological and biochemical functions of cold stress signal responses, but some of these genes have functions that have not yet been identified. Therefore, it is necessary to identify the physiological and biochemical functions of this pathway gene in rice and to study the response modes of cold stress signaling.

Due to the change in gene expression levels in the ICE-CBF-COR pathway, we analyzed the methylation level and found that most of the genes did not undergo methylation changes. However, *OST1*, which has three homologous genes in rice (*Os03g0610900*, *Os12g0586100* and *Os03g0764800*), One of the reported homologous genes, TRAR (*Os03g0610900*), is involved in regulating ABA signal transduction. Methylation analysis showed significant changes in the promoter region of this gene (Figure 10), the methylation level of which decreased after cold stress. qRT-PCR results showed that the gene was highly expressed, indicating that cold stress might cause demethylation of the promoter region and, hence, increase gene expression.

## 3. Discussion

### 3.1. The ICE-CBF-COR Conduction Pathway Plays a Leading Role in The Cold Tolerance of Varieties

Low temperature is an important environmental factor affecting plant growth and geographic distribution. OST1, a protein kinase, is a key component of plant resistance to low-temperature stress in the cold signaling pathway of the ICE-CBF-COR cascade. When OST1 kinase activity is activated by low temperature, OST1 positively influences the freezing resistance of plants by phosphorylating the key transcription factor ICE1 upstream of the CBF genes and stabilizing the ICE1 protein [9]. On the other hand, phosphorylation of the new polypeptide chain-coupled protein complex β subunit BTF3 indirectly regulates the stability of CBF protein and enhances the freezing resistance of plants [31]. Yang et al. (2019) identified the involvement of the PP2C E family protein phosphatase family EGR2 in the regulation of OST1 low-temperature activation [32].

In this study, the cold tolerance of rice seedlings was significantly positively correlated with the expression of ICE1 and the downstream CBF3 gene. The mechanism of ICE-CBF-COR signal transduction is speculated to occur via the demethylation of promoter cis-response elements induced by the upstream *OST1*-like gene, which may lead to the abnormal increase in gene expression. The signal is amplified in a step-by-step manner, and the signaling pathway is completed. After cold stress treatment, the expression levels of *OsHOS1(Os03g0737200)* and *OsDREB1G(Os02g0677300)* in the rice cultivar P427 increased significantly (Figure 10), indicating that the cold-tolerant pathway in P427, which was dependent on OST1 gene methylation, mediated the ICE-CBF-COR cold signal transduction pathway. The RNA-Seq analysis of the strong freeze-sensitive phenotype of the cbf1/cbf2/cbf3 triple mutant after cold acclimation showed that the CBF mutation affected 10–20% of the COR gene expression in the whole transcriptome [12,33]. CBF plays an important regulatory role in the low-temperature signaling pathway. The levels of cold tolerance of different rice varieties at the seedling stage are compatible with the climates of the producing areas, which are regulated by the CBF cold-responsive transcription pathway. In breeding programs, breeders can use the relevant gene expression information for the ICE1 regulation pathway as an auxiliary selection reference to breed new cold-tolerant rice varieties.

Although CBF plays an important role in cold-induced gene regulation, transcription factors in plants regulate the COR gene. Thoamshow et al. (2010) analyzed the RNA-Seq data of CBF-overexpressing lines and found that CBF2 and the transcription factors HSFC1, ZAT12, ZF, ZAT10 and CZF1 jointly regulate the downstream COR gene (there are cold-response genes, such as *GOL3*, that are completely dependent on CBF) [34]. Simultaneously, many studies have demonstrated the presence of cold-response genes that are independent of CBF in plants, indicating that the cold-response gene regulatory networks in plants are very complex and intrinsically linked; additional regulatory mechanisms remain to be further studied [12,35].

### 3.2. Association Analysis between DNA Methylation and Gene Expression

The gene promoter is a critical region for gene transcriptional regulation. Upon methylation of the promoter region, methyl groups and methylated proteins can inhibit the binding of transcription factors in the promoter region, leading to a decline or termination of gene transcription. Therefore, DNA methylation inhibits the expression of genes at the transcriptional level, and the level of gene expression affects the apparent characteristics of plants [8]. In this study, we further analyzed the methylation and expression levels of the promoter region of the main gene in the CBF transcription pathway regulated by *ICE1*. The results showed that the methylation degree of the *OST1* gene promoter region decreased after cold treatment in P427. The gene expression levels of *OST1*, *HOS1*, and *CBF3* also increased. The high expression of these genes further induced high expression of the cold-tolerant gene COR, leading to physiological and biochemical changes, as well as cold tolerance.

There are three main views concerning the effect of gene promoter methylation on transcription in plants. The first view is that promoter methylation inhibits transcription and affects gene expression [36]; the second view is that methylation of the promoter region does not affect transcription [37]; the third view is that DNA methylation in the promoter region can dynamically affect transcription [28,29]. Most findings support the first view that the hypermethylation level of the promoter region is highly inhibitory of the gene transcription level. The results of this study are consistent with the first view. We can speculate that the cold tolerance levels of different rice varieties are related to the ICE-CBF-COR cold signaling pathway, which regulates the transcription level of cold-response genes through the methylation level of the *OST1* promoter. The physiological response to low temperature ultimately leads to cold tolerance of different rice. This mechanism is very similar to the mechanism observed in *Eupatorium adenophorum*. However, the methylation variation of the *AaICE1* gene of *E. adenophorum* is mainly in the gene, not the promoter, region of rice [8].

### 3.3. Excavating New Gene-Rich Cold-Tolerant Regulatory Pathways

Low-temperature stress seriously affects plant growth and crop yield. To survive, plants have developed a complex and efficient regulatory network to resist and adapt to low-temperature stress, in which transcriptional regulation plays a key role. Transcription factors regulate the expression of a series of genes by binding to the cis-acting elements of the promoter region and play a key role in the plant abiotic stress response network. Transcription factors involved in regulating plant responses to low-temperature stress include AP2/ERF [38,39,40,41], MYB [42,43,44,45], Bhlh [46,47], NAC [48,49,50,51], ZFP [52,53], WRKY [54,55], etc. In this regulatory network, CBF transcription factors act as key molecular switches. CBFs specifically bind to the DRE/CRT cis-acting elements of the promoter region, thereby activating the expression of a series of low-temperature response genes, such as *COR*, *LTI*, *DHN*, and *RD*. The expression of *CBFs* is positively regulated by transcription factors, such as ICE1/2, CAA1, LHY, MYB56, ZFP1/182 and CAMTA1/2/3, but negatively by transcription factors, such as MYB15, MYBS3, ZAT12, PIF4/7, WRKY34 and EIN3 [10,56].

In this study, different rice varieties were analyzed after exposure to cold stress by MeDIP-seq, mRNA and Q-PCR, and differential genes that were sensitive to cold stress were screened and enriched in the KEGG pathway. Some new cold stress sensitive genes were found, e.g., *Os04g0497000*, annotated as Misc. oxidases-copper, flavone, etc. The expression profile of *Os04g0497000* was higher in P427 after cold stress than Nip and 9311 (Figure 8), indicating that the high expression of this gene was related to the cold stress resistance of rice. Aakash Chawade’s findings also showed that the greater the cold stress was, the higher was the expression of this gene [57]. However, the specific function of this gene in cold-tolerant regulatory pathways still requires follow-up studies.

Members of the WRKY transcription factor superfamily are required for the regulation of many plant pathways. In this study, the gene *WRKY70* was found to have a high expression level in the cold-tolerant variety P427 after cold stress. Zhang et al. showed that *WRKY70* inhibits Gibberellin (GA) and ABA signaling in a dose-dependent manner and that WRKY protein acts as a negative transcriptional regulator of GA and ABA signaling. These results suggest that the high expression of *WRKY70* contributes to the cold tolerance of plants [58,59,60].

In conclusion, the plant hypothermic response is a complex process that involves multiple genes and multiple signaling pathways. Methylation regulation and transcriptional regulation are only one aspect of the low-temperature response signaling network in plants. At present, research examining transcription factors has mainly focused on the transcription level, i.e., transcription factor binding to the cis-acting promoter element to activate or inhibit the expression of downstream target genes. Studies investigating post-transcriptional modifications of transcription factors, such as mRNA precursor splicing, editing, stability, nuclear transport, and small RNA-mediated mRNA degradation, are relatively lacking [39]. In future studies of plant hypothermic response mechanisms, priority should be given to elucidating the mechanism of action of posttranslational modifications in regulating the function of key transcription factors [61].

## 4. Materials and Methods

### 4.1. Low-Temperature Resistance Screening of P427

We selected 207 rice varieties used in agricultural production, including Indica and Japonica varieties, 50 seeds of each variety were planted in plastic pots. The culture conditions were as follows: day/night 28 °C/25 °C and 250 μM m^2^s^−1^ light. After growth of the plant to the three-leaf stage, it was transferred to an artificial climate chamber at 2–3 °C for 3 days; then, it was transferred to a light incubator for 7 days. According to phenotypic changes, screening for low-temperature-resistant varieties, variety P427 showed strong cold tolerance, Nip is more cold-resistant, 9311 cold-sensitive. Each sample was repeated three times.

### 4.2. Planting and Low-Temperature Treatment

P427 is a low-temperature-resistant variety that was obtained via low-temperature (4 °C) stress treatment of 207 local varieties and breeding materials. Japonica Nip and Indica rice 9311 were used as controls. Three seed materials were soaked at room temperature for 48 h and then germinated for 48 h at 32 °C. After germination was completed, the seeds were sown in a small plastic crucible and cultured in a light incubator. The culture conditions were as follows: day/night 28 °C/25 °C and 250 μM m^2^s^−1^ light. After growth of the plant to the three-leaf stage, it was transferred to an artificial climate chamber at 2–3 °C for 3 days; then, it was transferred to a light incubator for 7 days. The seedlings were cut with scissors before treatment, after treatment, and after growth, immediately wrapped in tin foil and stored in an ultralow temperature refrigerator at −80 °C for DNA and RNA extraction. This process was repeated 3 times.

### 4.3. Chlorophyll Content Determination

The method for determining the photosynthetic pigment content was carried out in accordance with the experimental method described by Porra [30]. In the three-leaf stage, three varieties were evaluated before treatment, after 3 days of low-temperature treatment, and after 1 day of growth. At these three different time points, the leaves were collected to measure the photosynthetic pigments. The specific method was as follows: 0.25 g of leaves were weighed, cut, placed in 30 mL of 80% acetone solution, and immersed in the dark at room temperature (25 °C) for 48 h, during which time shaking was repeated several times. After centrifugation for 10 min at 1000 r/m, the supernatant was aspirated, and the absorbance at 645 nm and 663 nm was measured using an HBS-1096 full-scale microplate reader. Each sample was averaged three times, and three independent biological replicates were evaluated for each sample. The specific formula for determining the chlorophyll content was as follows: Ca b = 20.3A645 − 8.04A663.

### 4.4. Hormone and Enzyme Content Determination

The phytohormone ABA and H_2_O_2_, O_2_^−^ content, SOD, POD, and CAT activities were determined by the Enzyme Linked Immuno Sorbent Assay (ELISA) referring to the measurement methods. The kit was purchased from Beijing Yonghui Biotechnology Co., Ltd. The test equipment was a PerkinElmer’s EnSpire Multilabel Reader 2300 microplate reader (PerkinElmer, PA, USA). At 9:00 in the morning, neat and uniform seedlings were used as test samples, for which 0.3 g of each sample was weighed and stored at −80 °C. This process was repeated 3 times.

### 4.5. RNA-Seq Assay

The sample materials of the three samples before and after low-temperature treatment were collected for RNA extraction, three independent biological replicates were evaluated for each sample, and RNA-Seq was determined by Shanghai OE Biotechnology Co., Ltd. Screening of differentially expressed genes was performed by analysis of GO Pathway (https://www.kegg.jp/), among which randomly selected genes were subjected to qRT-PCR verification.

### 4.6. MeDIP (Methylated DNA Immunoprecipitation Sequencing) Analysis

Total genomic DNA was extracted from rice seedlings grown to the three-leaf stage (day/night 28 °C/25 °C and 250 μmol m^2^s^−1^ light), and samples were then collected at a low temperature of 2–3 °C and normal temperature of 28 °C for three days using the Plant Dneasy Mini Kit (Qiagen, California, USA). The DNA purity was evaluated using a NanoPhotometer spectrophotometer (IMPLEN, CA, USA). The DNA concentration was measured using a Qubit^®^ DNA Assay Kit with the Qubit^®^ 2.0 Fluorometer (Life Technologies, CA, USA). The Methylated DNA Immunoprecipitation Sequencing was performed by Shanghai OE Biotechnology Co., Ltd. Three separate biological replicates were assessed.

### 4.7. Real-Time Quantitative PCR Validation of The DEGs

The RNA extraction kit from Novizan Biotechnology was used to extract the tissue samples from Nip, 9311, and P427 before treatment and after low-temperature treatment for 72 h. Then, the purified RNA was inverted into cDNA using HiScript@IIQ RT SuperMix supplied with the qPCR kit. The differential genes screened by the transcriptome were searched in the NCBI database (https://www.ncbi.nlm.nih.gov/), and qRT-PCR-specific primers were designed according to their sequences. The sample and internal reference gene ACTIN was evaluated 3 times each and the NTC (no template control) 2 times each. The quantitative expression analysis was performed by the quantitative PCR instrument with Bio-Rad CFX Manager V2.0 software. The 2^−∆∆*C*t^ method was applied for the measurements.

## 5. Conclusions

This study systematically analyzed the effects of cold stress on rice seedlings, as well as cold defense responses examined based on rice seedlings physiology, transcriptome, and methylation levels. Compared with Nip and 9311, a greater number of differentially expressed methylation and differentially expressed genes were observed in the P427 genotype. The low-temperature response of plants is a complex process that involves multiple genes and multiple signaling pathways. Methylation regulation and transcriptional regulation are just one aspect of the plant low-temperature response signaling network. The methylation and transcriptome analysis provided in this study extends our understanding of plant cold stress responses by identifying differentially expressed genes, such as transcription factors, signal transduction components, and genes involved in metabolism. This study provides a basis for understanding the cold tolerance mechanism of rice and for the development of engineering strategies to improve cold resistance in rice.

## Figures and Tables

**Figure 1 ijms-20-05089-f001:**
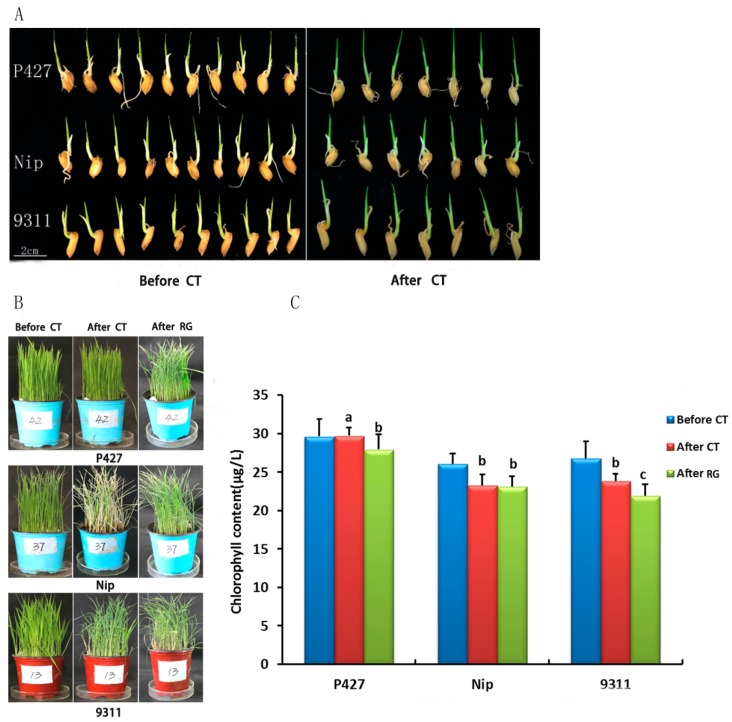
The Effects of Low-Temperature Stress on Seed Germination, Growth, and Survival of Three Rice Cultivars Seedlings. (**A**) Phenotype of chilling stress during rice germination. (**B**) Phenotype after chilling stress and after growth. (**C**) Phenotypic response to chilling in 9311, Nipponbare (Nip), and P427. Cold treatment (CT); Restore growth (RG). After CT, After RG, and Before CT, T test *a b c* represents *a* < 0.01, 0.01 < *b* < 0.05, *c* > 0.05.

**Figure 2 ijms-20-05089-f002:**
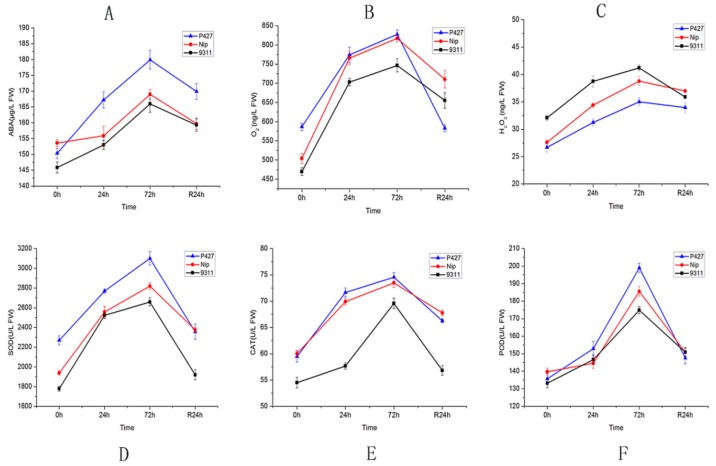
Physiological indices of the analyzed rice genotypes under chilling stress. (**A**) ABA content changes under chilling stress. (**B**) O_2_•^−^ activity changes under chilling stress. (**C**) H_2_O_2_ content changes under chilling stress. (**D**) SOD activity changes under chilling stress. (**E**) CAT activity changes under chilling stress. (**F**) POD activity changes under chilling stress.

**Figure 3 ijms-20-05089-f003:**
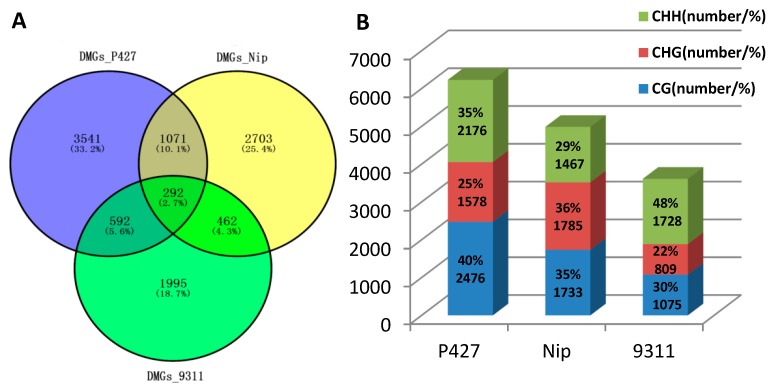
Comparison of DNA Methylation Levels and Loci. (**A**) Venn diagram of methylation genes in the promoter and gene body regions. (**B**) The ratio of methylation site (CHH, CHG, CG) in the promoter and gene body regions.

**Figure 4 ijms-20-05089-f004:**
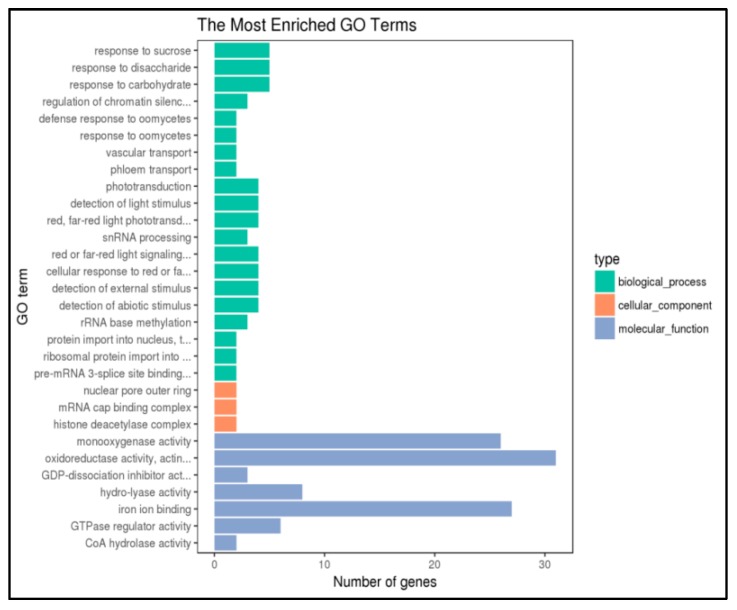
GO functional enrichment of differentially methylated genes in promoter regions of P427.

**Figure 5 ijms-20-05089-f005:**
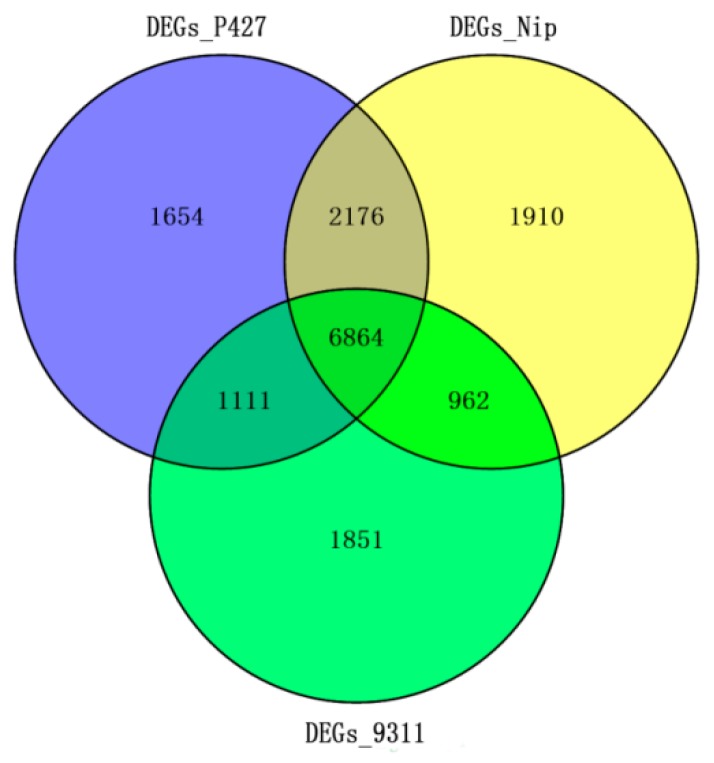
Venn diagram of differentially expressed genes (DEGs) in P427, NIP, and 9311.

**Figure 6 ijms-20-05089-f006:**
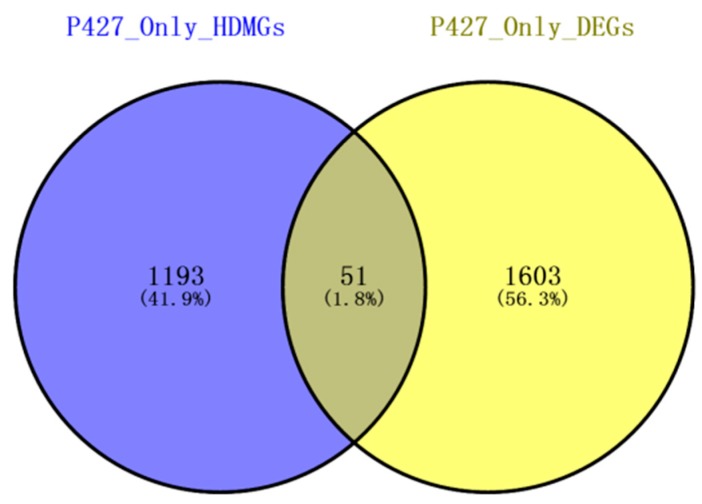
Diagram of P427-specific differentially expressed genes and hypermethylation differentially expressed genes in the promoter region.

**Figure 7 ijms-20-05089-f007:**
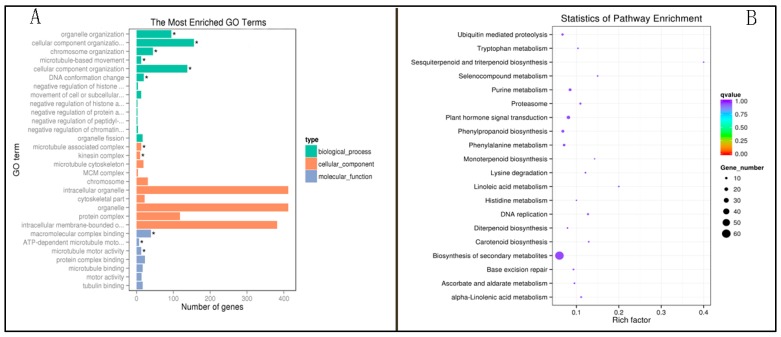
Gene ontology (GO) classifications of differentially expressed genes (DEGs) (**A**) Enriched GO terms for the 1603 genes (* *p* < 0.05). (**B**) Enriched KEGG pathways for the 1603 genes.

**Figure 8 ijms-20-05089-f008:**
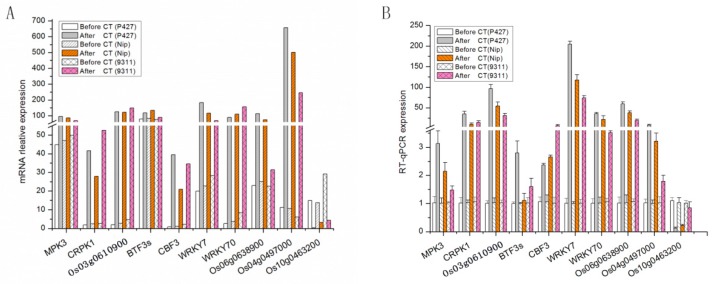
Comparison of gene expression levels using RNA-Seq and qRT-PCR. (**A**) Relative mRNA expression. (**B**) RT-qPCR expression. Relative expression values were normalized to the rice actin 1 gene. Error bars indicate the standard deviation.

**Figure 9 ijms-20-05089-f009:**
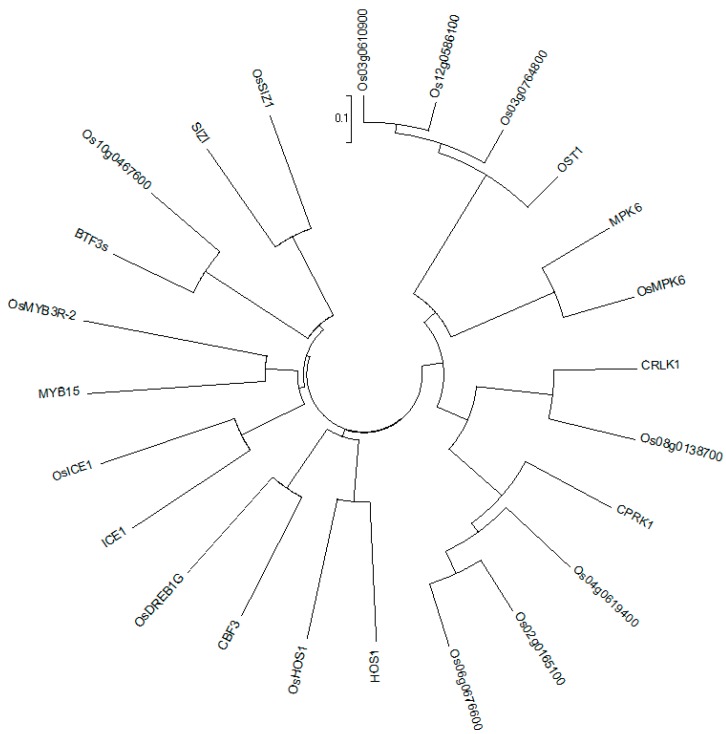
Homologous evolutionary tree of the ICE-CBF-COR cold stress response signaling pathway. The gene ID with Os is the rice gene, the other is Arabidopsis thaliana gene.

**Figure 10 ijms-20-05089-f010:**
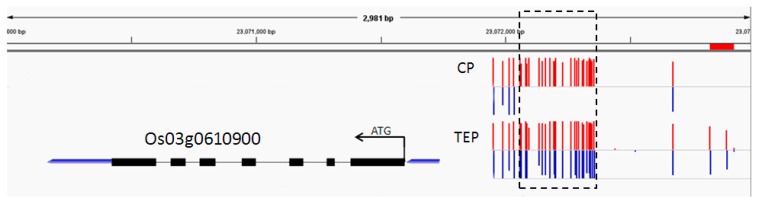
Comparison of the methylation degree of the promoter region of the *Os03g0610900* gene before and after cold stress. Red color indicates increased methylation levels, and blue indicates decreased methylation levels. CP: control group that did not undergo cryotherapy; TEP: experimental group that underwent cold treatment.

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
