# Peer review of "The Methylation Patterns and Transcriptional Responses to Chilling Stress at the Seedling Stage in Rice"

_ijms, 2019, doi:10.3390/ijms20205089_

Round 1

Reviewer 1 Report

The manuscript has described the methylation pattern and transcriptional alternation during cold stress.

there are several suggestions to help readers to understand easily these contents. 

  1. Add figure legend. Most figures, especially Figure 2, need more explanation. Figure title is a representative term, it’s can’t explain the content.
  2. Some data were missed statistical information. Add statistical analyzing results.
  3. What software, what method did you use for a phylogenetic tree? Did you use protein sequence or DNA sequence?
  4. In “Summary”, it seems some sentences to be generalized, even the final suggestion. For example, there is no any data or description to say “defense response” it may give confusion for readers to understand cold stress response.  Or, authors should check the term rice seedlings physiology rather than rice physiology.

Author Response

Dear editors

Greetings! According to the requirements of peer reviewers, I have revised the content and format of the manuscript submitted to IJMS, and answered the main questions as follows:

During the 2 months of the repair period, we have re-verified the physiological test, we can ensure that our physiological data is objective and reliable; RNA-seq raw data will be uploaded to the designated database website for public review, if editing Department requirements.

The reply to an editor's review comments is as follows:

  1. We have added legends and notes in the submitted manuscripts.
  2. In the newly submitted manuscript, we have added statistical methods and statistical analysis results.
  3. The authors used the protein sequence downloaded from the NCBI database, first using the clustalxdata analysis software for complete comparison (exemption pairing), and then constructing the phylogenetic tree through the MAGE7 bioanalytical software.
  4. Our conclusions are inferred from the existing experimental results. Of course, more experiments are needed to verify, in order to draw conclusions more accurately, and subsequent experiments areunderway. The imprecise wordsin this manuscripthas been revised.

Reviewer 2 Report

Analysis of the methylation patterns and transcriptional regulation of rice chilling stress response at seeding stage

by authors:

 Hui Guo , Tingkai Wu, Shuxing Li, Qiang He, Zhanlie Yang, Wuhan Zhang, Yu Gan, Pingyong Sun, Guanlun Xiang, Hongyu Zhang , Huafeng Deng

The presented manuscript is a revised version of a former work with decision: rejected.

The authors choose three varieties of rice according their cold stress tolerance: P427 (identified from earlier experiments as cold tolerant), Nip (Nipponbare, japonica rice, cold stress tolerant) and 9311 (indica, cold stress sensitive).

The authors present germination tests performed at low temperature with no observed difference.

Further they perform an experiment with 3 week old plants and stress them for 3 days with low temperature. For recovery plants were grown for 7 days at normal growth condition. The chlorophyll content and several biochemical parameters and enzyme activities (ABA, O2x, H2O2, SOD, CAT and POD) were estimated. Therefore samples were taken at time point 0, 24h, 72h, and 24h after regeneration.

DNA methylation analysis was performed from the three varieties by MeDIP seq and a transcriptomic analysis was performed.

General Comment:

Compared with the earlier version the manuscript improved.

Nevertheless there are several inconsistencies and points which are difficult to understand. Many of the points were raised earlier, but not addressed:

1. One main point is for example the analysis of the DEGs: In the method part line 505 the authors write: The RNA extraction kit from Novizan Biotechnology was used to extract the tissue samples from Nip, 9311 and P427 before treatment and after low-temperature treatment for 24 h and 72 h

Which time point was used to determine the DEGs?

  • 2. In line 122 the authors describe P427 as tolerant variety, but do not give detailed information about the selection process. Either the reference is lacking or the screening procedure must be described.
  • 3. From the presented picture (Figure 1A) it is not clear whether the yellow leaf tips are necrotic or senescent. In any case it cannot be concluded as PCD process ahead of time (line 142). Such statement would need further experiments (TUNEL stain, etc)
  • 4. Line 191: Together, these data showed that the decrease in CHH in rice after cold stress might be helpful in improving their cold resistance. This statement should be changed: this implies a functional molecular connection of cytosine methylation with cold stress resistance which was not followed up. Further figure 3b indicates the opposite: the cold tolerant line P427 has the highest amount of CG methylation and the level of CHH methylation is between Nip and 9311.
  • 5. Line194: To identify genes with significant differences in DNA methylation levels before and after cold stress, the genes with an expression change ≥ 2-fold (|log2Foldchange|≥1) and with a p value < 0.05 were defined as differentially methylated genes (DMGs): The authors refer to read abundance but not expression change. Must be changed.
  • 6. As I mentioned in the review before I would strongly recommend to the authors to include an analysis of the non-stressed control in their experimental setup. It would be important to see the difference between the lines under control conditions. Analyzing stress condition only, makes it difficult to impossible to address all meaningful differences.
  • 7. Line 194 ff: “In order to determine the genes with significant differences in DNA methylation levels before and after cold stress. Genes with an expression change ≥ 2-fold (〡log2Foldchange〡≥1) with P value < 0.05 were defined as differentially methylation genes (DMGs).” This sentence doesn’t make any sense. And I don’t understand what the authors want so say here. Expression change does not equal methylation change.
  • 8. Line 189: “modes of cytosine methylation CG, CHG, and CHH” this refers to methylation in different contexts and should be describes as such.
  • 9. Line 190ff: “are similar in proportion to these three materials (CG>CHH>CHG)” which material? And it is very unusual to find more CHH than CHG methylation. In several other publications the rate of CHG is described to be higher than CHH.
  • 10. Line496: In the method part the authors describe the MeDIP procedure:

line 502: The genome-wide bisulfite sequencing was performed by Beijing Nuohe Zhiyuan Technology Co., Ltd. Three separate biological replicates were assessed. I am becoming confused again…are they sure what they did?

  • 11. Line 193: “In order to determine the genes with significant differences in DNA methylation levels before and after cold stress.“ In the method part (line465) the authors describe only one condition 28°-25°C. I am completely confused now: What did they compare? Did they compare the varieties or the control vs stress condition?
  • 12. The experimental description of the DNA methylation analysis must be strongly improved. It is not clear whether a MeDIP (suppl table 2) or a WGBS sequencing approach was used.
  • 13. The ICE-CBF-COR pathway is very well investigated in Arabidopsis. It would be of importance to address the regulation via CBF1, CFB2 and CBF3. Why are the rice CBFs not included.
  • 14. Figure 8: Legend should be improved. Text: is the tree derived from AA sequence or nucleotide sequence? This is not a “Homologous evolutionary tree”, the depicted tree is not meaningful. Orthologues of CBF2 and CBF 1 are not investigated and should be included.
  • Also not all
  • Line 333 Methylation analysis: what was done here? The results look like BS-results… Info missing.
  • 15. Figure 10: Red indicates increased methylation levels, and blue indicates decreased methylation levels. Why are in TEP the same positions equally blue and red? Is it possible that this refers to top and bottom strand?

Decision reject 

Author Response

For the second editorial comment, the reply is as follows:

1. For the DGE analysis of this manuscript, we extracted and verified the RNA at 72h after cold treatment.

2. The screening process for P427 cold resistance has been added tothe newly submitted manuscript.

3. The TUNEL staining experiment has been supplemented. Although the results of the first experiment showed that pcd occurred in the mesophyll cells after low temperature treatment, the leaves were withered, but the staining effect of the picture was not clear enough. In order to get better experimental results, we The experiment is being repeated and this cycle may be a bit long. Prior to this, we have reached the above conclusions based on some physiological and biochemical data.

4. The way of presentation on line 191 has been modified;

5. The statement on line 194 has been corrected.

6. The 0h (ck)in our experiment is the non-stress control group under normal growth conditions.

7. Line 194, we mean the change in reading abundance readings of gene methylation;

8. Line 189, the description has been revised;

9. On line 190, the three materials we refer to are the three rice varieties used in the study, which refer to Nip, 9311, and P427, respectively. We performed methylation sequencing under cold stress, and CHH was more than CHG, which is the conclusion we obtained from objective analysis of sequencing data.

10. Methylation DNA immunoprecipitation sequencing was performed by Beijing Nuohe Zhiyuan Technology Co., Ltd., and thethree biological replicates we provided were measured separately. The description has been modified.

11. Line 93, the description is incorrect and has been revised.

12. The experimental description of DNA methylation analysis has been modified;

13. The CBF gene in Arabidopsis is highly homologous to the CBF gene in rice. The quantitative primers we designed are homologous sequences of CBF genes from two species;

14. This experiment uses the amino acid sequence for homologous phylogenetic tree analysis. We judge the orthologs of CBF1 and CBF2 based on the amino acid homology similarity;

15. Figure 10 refers to the situation where the methylation reading is up-regulated and down-regulated under normal conditions

Reviewer 3 Report

The authors took into account the comments made to the otiginal version of the manuscript and the article can be accepted for publishing after minor corrections. Few minor notes concerning text editing are mentioned in the enclosed file. Also I have two notes to the supplementary file:1)  note to the table S1 is uncorrect; 2) in the heading to the table S6 it is necessary to indicate what genes are kept in mind.

Author Response

Answer to the third editor as follows:

The suggestions made by the editor have been revised

Round 2

Reviewer 1 Report

  1. Please, clarify the statistical analysis. P>0.05 is not reliable?? What analyzing tool did you use? Add figure legends.
  2. For example, what are A, B C, etc in Fig.2? Increase figure 2 resolution.
  3. What’s the figure 3 title?
  4. At the conclusion of this study, authors demonstrated that methylation and gene expression is different during cold stress in rice. However, where is the basis to say “rice and other cereal crops” still this sentence is generalized. It seems that this sentence needs to be more revised.
  5. For the Figure 8, what sequence did you use? For saying methylation and gene expression, the evolutionary tree would be better with DNA sequence rather than protein sequence. In addition, this analyzing information at the figure legend. Add figure legends rather than use “note”.
  6. Describe what’s the content. Finished the sentence what supplementary materials (line 524)

Author Response

Dear Reviewer,

In order to make the research results more accurate, we have made partially revises in language. Secondly, we have made corresponding modifications to each suggestion in the manuscript of the reviewers. Here, we would like to express our gratitude to the reviewer for his Suggestions, which is very important for us to further improve the article.

1. Here the sentence description is not accurate, the author's expression means P> 0.05, the difference is not significant, using the T-text test method that comes with Excel.

2. Fig. 2 (A)ABA content changes under chilling stress.(B) O2•- activity changes under chilling stress.(C)H2O2 content changes under chilling stress. (D) SOD activity changes under chilling stress. (E) CAT activity changes under chilling stress. (F) POD activity changes under chilling stress. The author has made changes in the legend in the submitted manuscript;The resolution of Figure 2 has been modified in the submitted manuscript.

3. The figure 3 title is Comparison of DNA Methylation Levels and Loci; (A) Venn diagram of methylation genes in the promoter and gene body regions. (B) The ratio of methylation site (CHH, CHG, CG) in the promoter and gene body regions. The author has made changes in the submitted manuscripts.

4. The meaning of this sentence is: This paper studies the methylation and partial gene expression during cold stress of rice variety P427, which is helpful to understand the methylation and gene expression of other rice varieties under low temperature stress. It can also provide a theoretical reference for analyzing the methylation and gene expression of other cereal crops during low temperature stress. We have revised in the resubmitted manuscript to "this result provides the basis for further study of rice environmental stress, the relationship between DNA methylation and gene expression".

5. Figure 8 shows the evolutionary tree constructed by amino acid sequence. In the resubmitted manuscript, we reconstructed the evolutionary tree with DNA sequence, and found that it was consistent with the evolutionary tree constructed by amino acid alignment. The legend of Figure 8 has been modified in the submission manuscript.

6. The content of our description is that supplementary materials can be found in the attachment. The author has already made a revision in the resubmitted manuscript.

Reviewer 3 Report

All my comments have been taken by the authors into account in the revised manuscript. A few minor corrections can be made in the text: 

  1. Line 136 -RG? The legends on the left and right parts of the picture are different (After RG or After GR?). It is necessary to correct.
  2. Line 179 - Change indexes to indices Line 275 - Change up to increased Line 339 - Change arabidopsis to Arabidopsis Line 462 - Change comma to dot after pots

Author Response

Dear Reviewer,

In order to make the research results more accurate, we have made partially revises in language. Secondly, we have made corresponding modifications to each suggestion in the manuscript of the reviewers. Here, we would like to express our gratitude to the reviewer for his Suggestions, which is very important for us to further improve the article.

Response to Reviewer 2 Comments

1. This is the author's writing error and it has been revised in the submitted manuscript.

2. We have made the corresponding revision in the submitted  manuscript.